# Elevated-Temperature Tensile Properties of Low-Temperature HIP-Treated EBM-Built Ti-6Al-4V

**DOI:** 10.3390/ma15103624

**Published:** 2022-05-19

**Authors:** Karthikeyan Thalavai Pandian, Magnus Neikter, Fouzi Bahbou, Thomas Hansson, Robert Pederson

**Affiliations:** 1Department of Production Technology, University West, 461 86 Trollhattan, Sweden; magnus.neikter@hv.se (M.N.); thomas.hansson@hv.se (T.H.); robert.pederson@hv.se (R.P.); 2GE Additive, 435 33 Molnlycke, Sweden; fouzi.bahbou@ge.com; 3GKN Aerospace Engine Systems, 461 38 Trollhattan, Sweden

**Keywords:** additive manufacturing, electron beam melting, hot isostatic pressing, elevated temperature tensile strength, high-temperature mechanical properties, Ti-6Al-4V

## Abstract

Evaluation of the high-temperature tensile properties of Ti-6Al-4V manufactured by electron beam melting (EBM) and subjected to a low-temperature hot isostatic pressing (HIP) treatment (800 °C) was performed in this study. The high-temperature tensile properties of as-built and standard HIP-treated (920 °C) materials were studied for comparison. Metallurgical characterization of the as-built, HIP-treated materials was carried out to understand the effect of temperature on the microstructure. As the HIP treatments were performed below the β-transus temperature (995 °C for Ti-6Al-4V), no significant difference was observed in β grain width between the as-built and HIP-treated samples. The standard HIP-treated material measured about 1.4×–1.7× wider α laths than those in the modified HIP (low-temperature HIP)-treated and as-built samples. The standard HIP-treated material showed about a 10–14% lower yield strength than other tested materials. At 350 °C, the yield strength decreased to about 65% compared to the room-temperature strength for all tested specimens. An increase in ductility was observed at 150 °C compared to that at room temperature, but the values decreased between 150 and 350 °C because of the activation of different slip systems.

## 1. Introduction

Ti-6Al-4V is the most commonly used α+β titanium alloy in aerospace engine applications due to its high specific strength up to about 300 °C [1]. Normally, the alloy is manufactured as castings or forgings and then machined to the final geometry. These conventional manufacturing processes generate more waste material as compared to additive manufacturing (AM), which can produce more near-net-shape geometries directly from the feedstock [2,3]. Electron beam melting (EBM) is a powder bed fusion AM technique that has been widely researched in the past decade to better understand the microstructure formation in the EBM-built Ti-6Al-4V alloy [4,5,6]. The static mechanical properties of EBM-built Ti-6Al-4V components are comparable to their wrought counterparts [7,8]. The as-built EBM material can contain defects such as gas pores, which are detrimental to the fatigue properties [9]. It requires post-processing such as HIP treatment to produce nearly fully dense components [10].

At room temperature, the Ti-6Al-4V alloy comprises mainly of α phase (hexagonal close-packed crystal structure, HCP), with some remaining β phase (body-centered cubic crystal structure, BCC) [11]. The macroscopic plastic deformation behavior in conventional Ti-6Al-4V results from dislocation slip in both α and β phases [12]. In the case of EBM-built Ti-6Al-4V, the volume fraction of the β phase was reported in two separate studies to be about 2.7% [4,13]. Thus, the HCP slip systems will be primarily contributing to the dislocation mechanism in the EBM-built Ti-6Al-4V alloy. The dislocations in the HCP material can occur in basal, pyramidal, or prismatic planes [14]. Deformation by twining is suppressed in Ti-6Al-4V due to small phase dimensions and precipitates [12]. The dislocation gliding takes place on different slip planes depending on the HCP material’s temperature [15]. The elongation property of the material is therefore impacted by the activation of different slip systems at different temperatures [16].

Intertwined α laths separated with a thin layer of β result from moderate cooling rates and are called basketweave, which is the typical Ti-6Al-4V microstructure reported by many researchers for the EBM process [17,18,19]. The α laths are distributed within the elongated columnar prior β grains [20,21]. The width of prior β grains is primarily determined by the dwell time above the β transus temperature during solidification [22]. The prior β grain size remains unaffected when the post-heat treatment is performed below the β transus temperature [23].

For post-heat treatments below the β transus temperature down to 600 °C, the α laths were reported to coarsen [24]. HIP treatment is a commonly used post-heat-treatment process to remove pores in EBM-built Ti-6Al-4V and is typically performed at 920 °C with 100 MPa pressure, which results in a coarsening of the α laths [18]. It has been reported that the yield strength of a material HIP-treated at 920 °C with 100 MPa pressure is lower than that of the as-built material [5,25]. However, the reduced porosity by HIP treatment is believed to be beneficial for the ductility as the HIP-treated material has shown slightly increased elongation values than the as-built material has [25].

Lowering the HIP treatment temperature to 800 °C and increasing the pressure to 200 MPa has proven to close the pores with less impact on the yield strength [26]. Earlier work on the high-temperature behavior of conventional Ti-6Al-4V has shown a 35% reduction in yield strength around the maximum operating temperature of 300 °C [12,27]. A slight decrease in ductility is observed at around 250 to 300 °C, which is not a common trend in the elongation property at elevated temperatures for most other metals [16]. To the best of the authors’ knowledge, limited research has been conducted to study the effect of high temperature on the tensile properties of EBM-built Ti-6Al-4V. In addition, the impact of elevated temperature on the yield strength and elongation properties of a low-temperature HIP-treated material has not yet been investigated. In the present work, the tensile behavior at elevated temperatures of a low-temperature HIP-treated material is investigated and compared with the corresponding properties of as-built and standard HIP-treated material.

## 2. Materials and Methods

### 2.1. Material and HIP Treatment

The experimental samples were manufactured with plasma-atomized Ti-6Al-4V powder from AP&C (AP&C—a GE Additive Company, Saint-Eustache, Saint-Eustache, QC, Canada) with a particle size distribution of 45–105 µm and average diameter of 69 µm. The samples were built in an ARCAM Q20 plus EBM machine (GE Additive, Mölnlycke, Sweden) with a 5.3.76 process theme. Cylindrical samples of dimensions Ø15 mm × 105 mm long were machined along the build direction (BD) to Metcut drawing MRI 1512, as shown in Figure 1, for elevated-temperature tensile testing. For the room-temperature tensile test, the specimens were machined by GE Additive as per specimen 3 dimensions in Figure 8 in ASTM E8 standard [28]. Cubes with dimensions of 15 mm × 15 mm × 15 mm were manufactured with the same process theme and used for the metallographic study. The samples were evaluated in as-built and different HIP-treated conditions. The as-built samples were subjected to two different HIP treatments: standard HIP (920 °C, 100 MPa, 2 h hold time) and modified HIP (800 °C, 200 MPa, 2 h hold time).

### 2.2. Metallographic Investigation

The sample cross-sections were prepared using conventional lab instruments for cutting, mounting, grinding, and polishing. Before microstructural characterization, the samples were etched with Barker’s reagent for titanium (84 mL H_2_O + 16 mL 50% HBF4) by immersing the sample for about 10–12 s. The images for microstructural characterization were obtained using a light optical microscope (LOM) Zeiss AL10 (Zeiss, Oberkochen, Germany) and scanning electron microscope (SEM) ZEISS GeminiSEM 450 (Zeiss, Oberkochen, Germany) equipped with an electron backscattered diffraction (EBSD) detector Symmetry (Oxford Instruments, Abingdon, UK). An acceleration voltage of 15 kV, step size of 3 μm, and magnification of 150× were used to obtain sufficient interaction volume and broad area coverage. The α lath thickness measurement was performed on images obtained by LOM at 1000× magnification using the image analysis tool ImageJ 1.52a [29]. For α lath thickness measurement, images were taken at the top, bottom, left, right, and center of the cross-sections prepared parallel and perpendicular to the BD. The α lath width was measured by employing the interlamellar spacing measurement method provided by Ridley [29]. Multiple straight lines of known length are overlayed on the microstructure intersecting the α laths at different angles. The mean intercept spacing (l) is obtained by dividing the total line length by the number of intercepts. The true α lath spacing (λ_0_) is then calculated by dividing the mean intercept spacing by 2 [30].

The images for porosity measurement and inverse pole figure (IPF) maps were obtained from a polished surface. The average porosity distribution percentage was determined by taking images at 500× magnification at 35 locations in each cross-section and analyzing them using ImageJ. The relative density was calculated as the difference between a 100% dense sample and the calculated average porosity distribution percentage. It is difficult to distinguish clearly the prior β grain boundaries and measure the grain width in the macroscopic image. A more convenient method for obtaining the prior β grain width is by using inverse pole figure maps to reconstruct the β grains. The β grain maps were reconstructed based on the burgers orientation relationship (BOR) using a program to automatically reconstruct the parent grains from EBSD data called ARPGE 2.4 [31]. The β grain width was measured parallel and perpendicular to the BD using the mean line intercept as per the ASTM E112-13 standard [32]. The average intercept was determined by drawing horizontal lines at 10 locations in each reconstructed β grain image for all samples. The fracture surfaces were studied in the SEM at 500x magnification. The average dimple size on the fracture surface was measured from the SEM images using the intercept method.

### 2.3. Tensile Testing

The tensile testing was performed at four different temperatures of 20,150, 250, and 350 °C in air. The EBM-built and HIP-treated samples were machined and tested along BD at elevated temperature per ASTM E21 (17) [33] at Metcut Research Inc. in Cincinnati, OH, USA. The high-temperature test was carried out using a strain rate of 0.005 mm/mm/min through a 0.2% yield and a 1.6 mm/min head rate after that until failure. The specimens were tested at room temperature at GE Additive, Sweden, using a speed of 5 MPa/s until yield point, followed by 1.5 mm/min until failure. A minimum of one sample was tested per material condition.

## 3. Results

### 3.1. Microstructure and Porosity

Figure 2 shows macroscopic images of the as-built, standard HIP, and modified HIP samples perpendicular (Figure 2a–c) and parallel (Figure 2d–f) to the BD. The white markings drawn in Figure 2 depict the prior β grain boundaries. The prior β grains have a circular morphology perpendicular to the BD (see Figure 2a–c). The columnar prior β grains grow epitaxially over several build layers, as seen in Figure 2d–f. The prior β grains are similar in size in both as-built and HIP-treated conditions. The images along the BD show that the prior β grains are inclined toward the ends of the sample, which implies that they tend to orient following the temperature gradient. Larger pores are only visible in the as-built sample (Figure 2a,d). In the HIP-treated samples, only smaller pores are found, and they are difficult to distinguish because of the low-magnification macroscopic images.

The inverse pole figure (IPF) maps and the respective reconstructed β grain images are taken along the BD and are shown in Figure 3 for the as-built, standard HIP, and modified HIP samples. The images reconstructed with β grains from the IPF using BOR are shown in Figure 3d–f. The images indicate that the β grains grow epitaxially in a columnar manner along the BD, similar to the observations of prior β grains in the LOM images (Figure 2d–f).

Table 1 shows the β grain width measurement using the intercept method as per the ASTM E112-13 [32] standard for the different material conditions. The quantitative data show that the HIP-treated materials’ average β grain width is similar to that of the as-built samples, indicating that the β grain width is unaffected by the different HIP treatments.

Figure 4 shows the microstructure of the as-built, standard, and modified HIP-treated samples perpendicular (Figure 4a–c) and parallel (Figure 4d–f) to the BD. The black markings in Figure 4a–c represent α laths. The white markings drawn in Figure 4d–f indicate grain boundary α (GB-α). The GB-α is mostly discontinuous in all three material conditions. Qualitatively, the α laths appear coarser in the standard HIP condition. In contrast, the modified HIP material and the as-built material have relatively fine α laths of similar size.

The α lath thickness measurements are presented in Table 2. The average α lath thickness is different in all three conditions and ranges from 0.84 to 1.54 μm along the BD and 0.97 to 1.53 μm perpendicular to the BD. Here, it is observed that the as-built sample has a smaller thickness than the HIP-treated samples. When comparing the different HIP-treated samples, the α lath thickness is larger in the standard HIP-treated material. The average α lath thickness (*δ_α-lath_*) utilized to calculate the theoretical yield strength using the Hall–Petch relation in Section 4.2 (Equation (3)) is provided in Table 2.

Example images for each of the three material conditions used for the section porosity analysis are shown in Figure 5. As seen from Figure 2a,d and 5a, the as-built condition contains larger pores. Figure 5b,c show that both the HIP-treated conditions have comparatively smaller pores.

The average porosity area and relative density, measured across the cross-section perpendicular to the BD for the three different sample conditions, are presented in Table 3. The as-built material shows the highest average porosity. There is no significant difference in the porosity values between the standard and modified HIP materials. The density for a theoretically fully dense material is assumed to be 100%. The relative density of the material is calculated and reported as a difference between porosity area % to the density of theoretically fully dense material. Consequently, the as-built material with the highest porosity shows the lowest relative density, while the HIP-treated materials have higher relative density values.

### 3.2. Tensile Properties

Figure 6 shows the variation of 0.2% yield strength with temperature for the three different materials. All three materials show maximum yield strength at room temperature, and the strengths are reduced with increasing test temperature. The yield strength at 20 °C is 930 MPa, 855 MPa, and 910 MPa, for the as-built, standard, and modified HIP materials, respectively. At 20 °C, the theoretical yield strength calculated based on the empirical Hall–Petch relation in Section 4.2 (Equation (3)) is plotted as 887 MPa, 853 MPa, and 876 MPa for the respective materials. The yield strength gradually reduces to 585 MPa, 505 MPa, and 573 MPa at 350 °C for the as-built, standard, and modified HIP conditions. At the same test temperature, the standard HIP specimen shows the lowest yield strength value (855 MPa at 20 °C and 505 MPa at 350 °C), whereas the as-built material shows the highest value (930 MPa at 20 °C and 585 MPa at 350 °C).

The effect of temperature on the ultimate tensile strength (UTS) for the tested specimens is shown in the UTS vs. temperature plot in Figure 6. Similar to the yield strength, the ultimate tensile strength decreases with the temperature. The maximum tensile strength is reported at 20 °C, and the values are 1002 MPa, 989 MPa, and 967 MPa for the as-built, modified, and standard HIP conditions. At 350 °C, the tensile strength is reduced to 725 MPa, 715 MPa, and 685 MPa for the respective material conditions mentioned earlier. At intermediate temperatures, the tensile strength gradually decreases to 835 MPa, 835 MPa, and 800 MPa at 150 °C and 750 MPa, 760 MPa, and 715 MPa at 250 °C for the as-built, modified, and standard HIP materials.

The 4d elongation % plot for the tested specimens in the 20 to 350 °C range is shown in Figure 6. The elongation % values at 20 °C are 17%, 18%, and 19% for the as-built, standard, and modified HIP materials, respectively. The elongation values increase to 19%, 22%, and 22% at 150 °C and then decrease to 17%, 21%, and 19% at 350 °C for the as-built, standard, and modified HIP conditions, respectively. It is observed that the standard HIP specimens have the maximum ductility in the high-temperature range. The as-built specimens have the lowest elongation values in the tested temperature range. The modified HIP specimens have ductility at the mid-level compared to the other tested specimens in the high-temperature regime.

### 3.3. Fractography

The SEM images of the fracture surfaces obtained from the tensile test bars are shown in Figure 7. Larger pores are visible on the fracture surfaces of the as-built material (see Figure 7a–c). The pores are less visible in the HIP materials (see Figure 7d–i). Qualitatively, the size of the dimples appears to increase with the test temperature for the as-built and HIP-treated material conditions. The dimples are smaller in size for the specimens tested at 150 °C (Figure 7a,d,g) than the dimples observed in specimens tested at 350 °C (Figure 7c,f,i). Intermediate-size dimples are observed in Figure 7b,e,h, which belong to the specimens tested at 250 °C.

Quantitative measurements of the average dimple size using the intercept method for the different test temperatures and material conditions are presented in Table 4. The effect of the test temperature on the dimple size can be inferred from the data shown in Table 4. The average dimple size increases with the test temperature for all material conditions. The average dimple size measured on the fracture surfaces of both as-built and HIP-treated material tested at 150 °C ranges from 3.9 to 4.4 μm. For the specimens tested at 250 °C, the dimple sizes are between 5.1 and 5.4 μm. In the case of as-built and HIP-treated materials tested at 350 °C, the average dimple sizes are measured in the range of 6.6 to 7.4 μm. The standard HIP-treated material shows a larger dimple size than the other material conditions tested at the same temperature.

## 4. Discussion

### 4.1. Microstructure and Porosity

In EBM-built Ti-6Al-4V, β grains generally have an epitaxial growth along the build direction [4,6], which is also observed in the present work (Figure 2d–f). The β grain width is not affected by the HIP treatments, because the HIP-treated materials are not exposed to temperatures above the β transus temperature [25,34]. In the present work, the average β grain width is 51 to 66 μm, in both as-built and HIP conditions. Therefore, no significant influence from the post-heat treatments on the prior β grains is observed in the present study as well. Larger prior β grain widths have been reported in the past for EBM-built Ti-6Al-4V [34,35]. Measuring the prior β grain widths using macroscopic LOM images can be subjective as one has to estimate where prior β grain boundaries are located. Other factors contributing to the uncertainty in such quantitative measurements are sampling size and image location [6].

During cooling down of Ti-6Al-4V, the β → α phase transformation starts at the β transus temperature of 995 °C [12]. Below the β transus temperature, the α laths typically nucleate from the β grain boundary and, depending on cooling rate, the α laths grow either along the grain boundary or into the β grains [35]. In the EBM process, Ti-6Al-4V cools down fast in the beginning at about 10^3^–10^5^ K/s to the build chamber temperature of 650–700 °C [5,24]. After the build is completed, the material slowly cools down to room temperature. Several publications relate the α lath thickness in Ti-6Al-4V to the cooling rate during the β → α transformation [4,12,25,35,36]. The β → α transformation in EBM can occur either as a diffusion-controlled process or as a martensitic transformation [4]. In the diffusion-controlled process, the β phase transforms into Widmanstätten α laths and GB-α during the rapid cooling from the β transus temperature down to the build chamber temperature [4]. The presence of GB-α in the microstructure of the investigated materials indicates that the transformation follows the diffusion-controlled process.

Other factors affecting the α lath thickness apart from cooling rate are the post-heat treatments. The post-heat treatment of the samples investigated here is performed at two different temperatures, with the same holding time and cooling rate. In the present study, the effect of HIP temperature on the α lath thickness is evident. The α laths coarsen with the increase in post-heat treatment temperature, which is similar to findings from previous studies [18,24]. The as-built material has the minimum α lath thickness, whereas the standard HIP material shows the maximum α lath thickness. It is clear from the results in Table 2 that the post-treatment temperature has a significant influence on the α lath size. Material subjected to the higher HIP treatment temperature of 920 °C results in an almost 1.7× increase in average α lath thickness than that in the as-built condition, while the low-temperature HIP treatment (800 °C) increases the average α lath thickness by only about 1.2×.

The quantitative results in Table 3 show that as-built samples have an almost five times larger porosity distribution than the HIP samples. Still, the larger pores in the as-built condition are less detrimental to the yield strength of the material as the load-bearing capacity of the material may not be affected by the presence of these pores. The as-built material has a higher yield strength than the HIP-treated samples in the tested temperature regime. On the other hand, the larger pores impact ductility due to strain localization around the pores, limiting the extent of uniform plastic strain before fracture. The as-built samples have large circular pores and an elongated lack of fusion defects. As a result, the stress concentration around such defects would be higher than the regions in HIP-treated samples with lesser porosity. The high stress concentration makes the strain in the section with large pores reach the fracture strain even at a lower macroscopic strain in the gauge section, resulting in earlier fracture initiation [37]. Therefore, the fracture starts around the large pore vicinity and grows to the surface much faster than the less porous sections do, leading to limited plastic deformation of the gauge section and overall elongation.

Even though the modified HIP treatment is performed at a lower temperature than the standard HIP, there is no significant difference in the average porosity % distribution. The increased pressure value in the modified HIP condition would have been beneficial to close the pores and obtain partial densification comparable to the standard HIP treatment. Both HIP treatments produce almost fully dense material.

### 4.2. Effect of Temperature on Tensile Properties

Reductions in room-temperature yield strength of about 20%, 30%, and 40% are observed at 150 °C, 250 °C, and 350 °C, for all three material conditions. The work of Conrad et al. [38] has shown that at temperatures < 0.4 *T_m_*, where *T_m_*—melting temperature, the yield stress of the polycrystalline single-phase titanium can be split into the thermal component (*σ**) and athermal component (*σ_G_*); see Equation (1).
(1)$$\sigma =\sigma^{*}\;\left(T,\;έ,\;C_{i}\right)+\sigma_{G\;}\left(G,\;\varepsilon ,\;d,\;C_{i},\;C_{s}\right)\;$$
where *T*—testing temperature, *έ*—strain rate, *C_i_*—interstitial solute content, *G*—shear modulus, *ε*—plastic strain, *d*—average grain size, and *C_s_*—substitutional solute content. The test temperature primarily affects the thermal component of the yield stress. Gysler et al. [39] further investigated the effect of test temperature for Ti-6Al-4V alloy and found that the phase dimensions did not significantly change the thermal component of the yield stress. Within the tested temperature range, the thermal component *σ** was observed to decrease linearly with the square root of the test temperature and proportionally to the interstitial solute content [38]. At room temperature, the interstitial impurities in the alloy hinder the dislocation movement and demand higher stress to initiate plastic deformations. An increase in test temperature induces thermally activated dislocation motion to overcome the interstitial barriers, which causes a rapid decrease in the yield strength.

In the present study, the standard HIP samples have about 10–14% lower yield strength than the as-built and modified HIP samples at the tested temperature range. The standard HIP sample has the thickest α laths among the tested materials, which could be one reason for the lowest yield strength. Hall and Petch have extensively investigated the effect of grain size on the polycrystalline material’s yield strength [40,41]. They found that as the average grain size increases, the yield strength of the material decreases. The rationale is that as the grains grow larger, the grain boundary area reduces, providing reduced resistance to dislocation movement and therefore initiating plastic deformation at lower stress. The generic Hall–Petch equation relating material yield strength (*σ_y_*) to average grain size (*d*) is shown in Equation (2) [42].
(2)$$\sigma_{y}=\sigma_{o}+k_{y}d^{-1/2}$$
where *σ_o_* and *k_y_* are material constants. An empirical relation of the Hall–Petch equation with EBM-built Ti-6Al-4V material constants at room temperature relating the yield strength to average α lath thickness (*δ_α-lath_*) is shown in Equation (3) [24]. The experimental values from the current study for the room-temperature yield strength are in close agreement with the yield strength calculated based on the measured average α lath thickness using Equation (3).
(3)$$\sigma_{y}=737+144\;\delta_{\alpha -lath}^{-1/2}\;$$

From Equation (1), it is evident that the average grain size impacts the athermal component of the yield stress. The room-temperature Hall–Petch relation can be applied for high-temperature deformations as the same type of highly planar dislocations are found to occur both at room temperature and at test temperatures up to 500 °C [39]. Consequently, the difference in yield strength based on the average α lath thickness observed at room temperature remains constant irrespective of test temperature.

The elongation values increase by about 10 to 20% between room temperature and 150 °C test conditions. Between 150 and 250 °C, the values decrease by about 5% for the modified and standard HIP materials, whereas there is no change in ductility in the as-built samples. In the 250–350 °C regime, the ductility drops by about 7 to 11% in the modified HIP and as-built materials. No change in elongation values is observed for the standard HIP materials in the last temperature regime. The decrease in ductility with an increase in temperature can be attributed to the influence of temperature on the activation of different deformation slip systems. At room temperature, <a> type slips are dominant dislocations on the prismatic plane [15]. In the temperature range of 150–300 °C, <c + a> modes on the pyramidal planes are more prominent, resulting in the decrease in elongation values [16]. The impact on ductility at elevated temperatures due to the activation of different slip systems will be further investigated in the future work.

The dimples that appear on the SEM images on the fracture surfaces are clear indications of ductile failure for all material conditions. The ductile fracture mechanism involves void nucleation, followed by void growth and coalescence, leading to rupture and the formation of dimples on the fractured surface [39]. The test temperature and average grain size both have an impact on the dimple morphology. An increasing trend in the dimple size is observed for both as-built and HIP-treated material conditions as the test temperature increases from 150 °C to 350 °C. As the test temperature increases, more void growth occurs preferentially than void nucleation, resulting in larger-sized dimples on the fracture surface [43]. The fracture surface of the standard HIP material, with the thickest α laths, has the largest average dimple size. Coarse α laths have a limited lath interface for the voids to nucleate, and with plastic deformation, the void growth is promoted over nucleation, resulting in large size dimples.

## 5. Conclusions

In this study, the elevated-temperature tensile properties of EBM-built Ti-6Al-4V material in three different conditions were investigated, namely in an as-built condition, with low-temperature HIP-treatment, and with standard HIP-treatment. The following conclusions were made from this work.

No difference in prior β grain size was observed between the three different materials.The smallest α lath thickness was found in the as-built material, while the α lath thickness increased with post-HIP-treatment temperature.Following the Hall–Petch relationship, the α lath thickness impacted the yield strength of the material. The standard HIP-treated material with maximum α lath thickness had about a 10–14% lower yield strength than the as-built and modified HIP-treated counterparts.A reduction in yield strength with increased temperature was observed for all three material conditions. At 350 °C, the yield strength dropped to about 37–41% of the corresponding room-temperature yield strengths, respectively.Compared with ductility at room temperature, the elongation % value increased by 2–4% at 150 °C, followed by a decrease of 1–3% at 350 °C.The yield strength was not affected by the porosity, but the porosity negatively impacted the ductility. The as-built material, with larger pores, had the maximum yield strength and minimum ductility.The fracture surfaces had dimples and their sizes varied with test temperature and α lath thickness. The specimens tensile-tested at 350 °C showed a 1.7x larger dimple size than those found in specimens tensile-tested at 150 °C. The material with the widest α lath size showed the largest dimple sizes.

## Figures and Tables

**Figure 1 materials-15-03624-f001:**
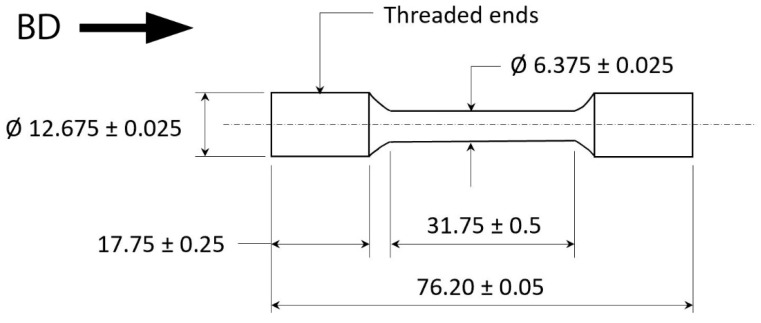
Tensile test specimen geometry. Unless otherwise specified, all dimensions are in mm.

**Figure 2 materials-15-03624-f002:**
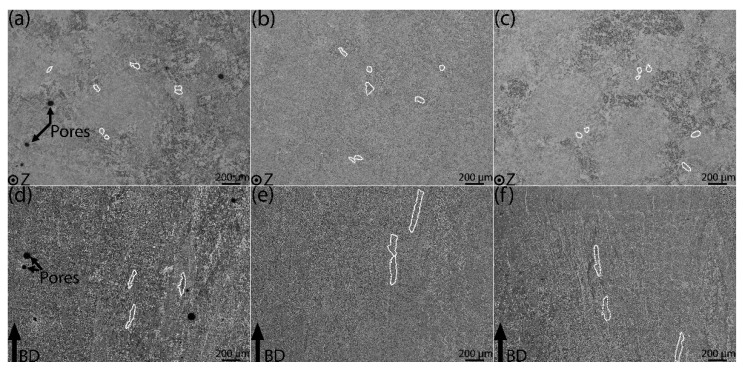
LOM images of EBM-built Ti-6Al-4V perpendicular (**top row**) and parallel (**bottom row**) to build direction (BD). Images (**a**,**d**) represent the as-built samples, images (**b**,**e**) represent the standard HIP samples, and images (**c**,**f**) represent the modified HIP samples. The white markings represent prior β grain boundaries.

**Figure 3 materials-15-03624-f003:**
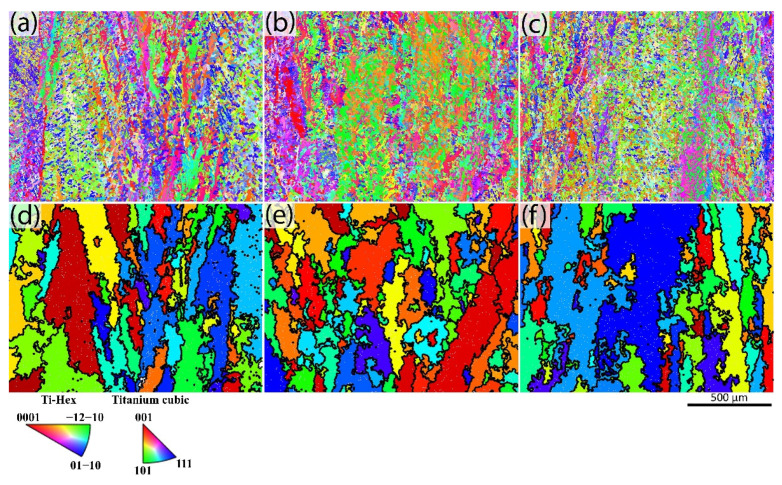
Inverse pole figure maps (**top row**) along BD obtained from SEM using EBSD and respective reconstructed β grain (**bottom row**) obtained from ARPGE software. As-built condition (**a**,**d**), standard HIP condition (**b**,**e**), and modified HIP condition (**c**,**f**).

**Figure 4 materials-15-03624-f004:**
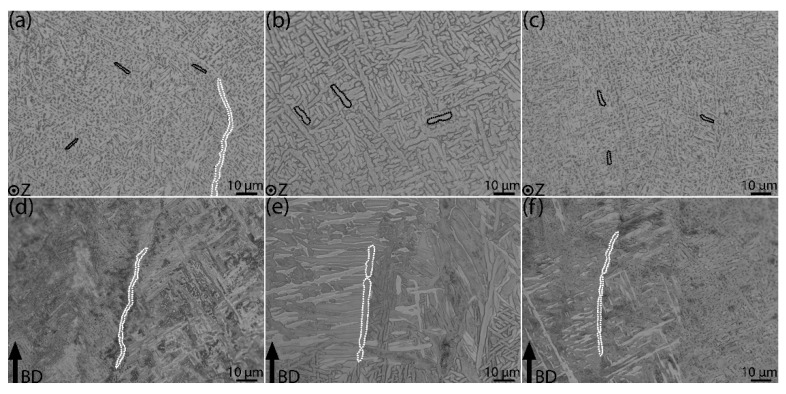
LOM images showing microstructure of EBM-built Ti-6Al-4V perpendicular (**top row**) and parallel (**bottom row**) to build direction (BD). Images (**a**,**d**) show the as-built microstructure, images (**b**,**e**) show the standard HIP microstructure, and images (**c**,**f**) show the modified HIP microstructure. The black markings represent α laths, while the white markings indicate grain boundary α (GB-α).

**Figure 5 materials-15-03624-f005:**
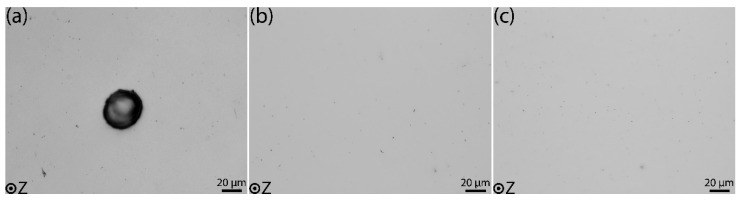
LOM images used for porosity measurement perpendicular to BD: (**a**) as-built condition, (**b**) standard HIP condition, (**c**) modified HIP condition.

**Figure 6 materials-15-03624-f006:**
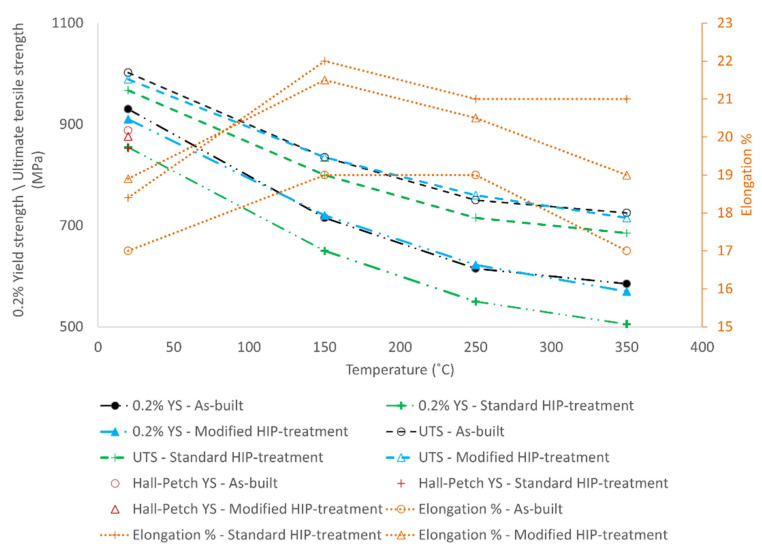
Tensile property plots for as-built and HIP-treated materials showing the variation of 0.2% Yield strength (YS), Ultimate tensile strength (UTS), and 4d Elongation % with temperature. At 20 °C, the yield strength calculated by the empirical Hall–Petch relation [24] using the average α lath thickness (*δ_α-lath_*) is plotted for the investigated material.

**Figure 7 materials-15-03624-f007:**
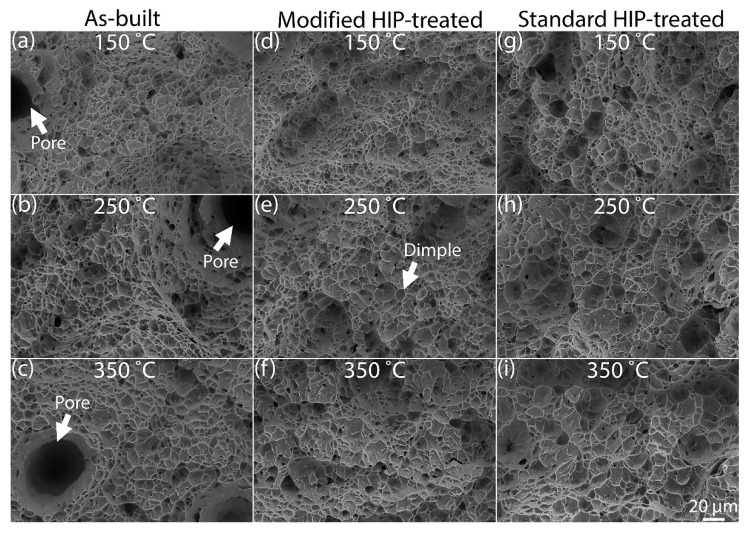
SEM images of fracture surfaces of the tensile test specimens. Images of fractured surfaces in as-built condition tested at 150 °C (**a**), 250 °C (**b**), and 350 °C (**c**) showing dimples and larger pores. Images of fractured surfaces in modified HIP-treated condition tested at 150 °C (**d**), 250 °C (**e**), and 350 °C (**f**) and standard HIP condition tested at 150 °C (**g**), 250 °C (**h**), and 350 °C (**i**) showing smaller pores.

**Table 1 materials-15-03624-t001:** Summary of average β grain width measured on the reconstructed images as per ASTM E112-13 [32] standard by intercept method.

Sample Description	Reconstructed β Grain Average Intercept Length (μm)	
	Parallel to BD	Perpendicular to BD
As-built	66 ± 15	51± 9
Standard HIP (920 °C, 100 MPa, 2 h)	58 ± 17	59 ± 14
Modified HIP (800 °C, 200 MPa, 2 h)	58 ± 11	57 ± 9

**Table 2 materials-15-03624-t002:** Average and standard deviation of α lath thickness for as-built and HIP-treated samples measured on images taken parallel and perpendicular to BD.

Sample Description	Average α-Lath Thickness (μm)		
	Parallel to BD	Perpendicular to BD	Average Thickness (*δ_α-lath_*)
As-built	0.84 ± 0.10	0.97 ± 0.14	0.91
Standard HIP (920 °C, 100 MPa, 2 h)	1.54 ± 0.28	1.53 ± 0.23	1.54
Modified HIP (800 °C, 200 MPa, 2 h)	1.12 ± 0.14	1.01 ± 0.11	1.07

**Table 3 materials-15-03624-t003:** Average porosity area (%) and the relative density (%) perpendicular to BD for as-built and HIP conditions.

Sample Description	Average Porosity Area %	Relative Density %
As-built	1.15 ± 1.98	98.85
Standard HIP (920 °C, 100 MPa, 2 h)	0.16 ± 0.12	99.84
Modified HIP (800 °C, 200 MPa, 2 h)	0.19 ± 0.06	99.81

**Table 4 materials-15-03624-t004:** Effect of test temperature on the dimple size formed on the fracture surface for the as-built and HIP-treated materials measured by intercept method.

Sample Description	Average Dimple Size (μm) on Fracture Surfaces for the Different Materials Tested at Different Temperature		
	150 °C	250 °C	350 °C
As-built	3.9 ± 0.43	5.1 ± 0.71	6.6 ± 1.18
Modified HIP (800 °C, 200 MPa, 2 h)	3.8 ± 0.28	5.3 ± 0.47	6.6 ± 0.59
Standard HIP (920 °C, 100 MPa, 2 h)	4.4 ± 0.38	5.4 ± 0.46	7.4 ± 1.14

## Data Availability

The data used to produce the results in this study are also part of another ongoing investigation. Therefore, the data cannot be shared at this point of time.

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
