# Peer review of "Elevated-Temperature Tensile Properties of Low-Temperature HIP-Treated EBM-Built Ti-6Al-4V"

_materials, 2022, doi:10.3390/ma15103624_

Round 1
Reviewer 1 Report
Karthikeyan Thalavai Pandian et al., elaborated the paper entitled “Elevated temperature tensile properties of low-temperature HIP treated EBM built Ti-6Al-4V”, where compressive studies regarding Ti-6Al-4V manufactured by electron beam melting (EBM) and subjected to a low-temperature hot isostatic pressing (HIP) treatment were conducted and followed by consistent experimental findings and well disseminated discussion chapter. The manuscript has a well-organized and easy to follow structure, without any noticeable grammar errors. Minor changes to the manuscript should be addressed as following:
- Affiliation numbering should be revised;
- If possible, authors should avoid using very old references;
- Authors should state clearly was is the novelty of the proposed research;
- Manufacturer information for used apparatus and materials should be mentioned as required in the manuscript preparation instructions; Manuscript format should be revised;
- Figure 1 should contain plane coordinates for better understanding;
- Ti-6Al-4V powder characteristics should be added (e.g., diameter range, purity etc.);
- Please add details or reference regarding “EBM machine with a 5.3.76 process theme” paragraph;
- In material and methods chapter, authors should insert a table containing detailed information regarding the studied samples;
- Related to macroscopic findings in figure 2, authors should introduce a threshold for categorizing the pore dimensions; Adding a quantitative pore statistic could be useful;
- Only as a suggestion, the IPF maps should be correlated with the macroscopic images (i.e., scale, region on measurement etc.);
- In table 1 and 2, authors should reconsider adding to “sample description” column the parameters (temp., pressure, time etc.), in order to avoid confusion with the term HIP-treated materials;
- Figure 5 should be assisted with an overview image for standard and modified HIP samples in order to emphasize the lack of pores for extended regions;
- As mentioned for figure 2, in figure 7 one should state the observed pore diameters; Consider adding pore dimensions directly on SEM images; Also, dimple observations should be highlighted by magnified images;
- Conclusion section should include future foreseen research based on the current findings.
Reviewer 2 Report
This paper is an interesting report on the evaluation of high-temperature tensile properties of Ti6Al4V produced by EBM and subjected to a low-temperature HIP treatment. This work may be accepted if the authors can address the following issues:
- 1 is not necessarily required and writing the ASTM code is enough.
- In Fig. 3 the phase map of EBSD analysis should be added.
- 5 is not presenting the meaningful data and should be replaced with a clearer SEM image and EDS mapping.
- Captions of all figures are not informative enough and should be explained in more detail like information on image type (SEM or optical microscope).
- XRD of investigated samples in different steps of the process should be added.
- The stress-strain curves of the tensile test should be added to the manuscript or as a supplementary file.
- The reasons for porosity's effects on strength and ductility must be well described.
- The author should provide more information about starting materials (powder), such as the average and distribution of particle size, or even add SEM powder in the manuscript or supplementary.
Reviewer 3 Report
The manuscript is of scientific interest and importance. New data connected with the influence of different HIP conditions on microstructure and properties of EBM Ti-6Al-4V alloys are provided.
However, there are some points should be added and corrected.
Line 84. There should be placed the reference to the ASTM E8 standart.
Line 120. Abbreviation ARPGE should be transcribed.
Line 121. There shpuld be added the reference to ASTM E112-13 standart.
Line 129. The same for ASTM E21 (17) standart.
It is unclear was the atmosphere of tensile testing process controlled? If it was, what atmosphere was used? Specially this should be specified for the tensile tests carried out at different temperatures.
Author describe alpha+beta phase state according to the optical and SEM results. However, there is no information confirming the phase state. There should be provided results of EDS analysis (alpha and beta phases have different elemental compositions and different concentrations of Ti whithin the grains) or XRD (phase analysis confirms alpha+beta state) or TEM analysis (SAED patterns could confirm presence of alpha and beta phases). This should be clarified by at least one method.
It is unclear what was the volume fraction of beta- and alpha-phases' grains in the studied samples.
Fig. 6. There should be added the standart deviation for each point on the plot.
Line 287. I guess, there should be 10^3-10^5 K/s instead of 103-105.
It is unclear, which orientation of EBM-material was chosen for the formation of samples for tensile tests. This should be clarified.
Round 2
Reviewer 2 Report
The authors did not implement all of the changes I requested. According to the previous comments, the manuscript still needs to be improved.
Author Response
Hi,
Thank you for reviewing the manuscript and providing your valuable feedback. I have implemented most of the changes suggested in the round 1 review. I have also mentioned the reason for not implementing the other changes. Please reconsider my comments and provide your feedback. Thanks again for your time.
Reviewer 3 Report
The manuscript was significantly improved and can be published.
Author Response
Hi,
Thank you for your time in reviewing the manuscript and for providing your valuable feedback. Much appreciated!